# A Study on the Endangerment of *Luminitzera littorea* (Jack) Voigt in China Based on Its Global Potential Suitable Areas

**DOI:** 10.3390/plants14172792

**Published:** 2025-09-05

**Authors:** Lin Sun, Zerui Li, Liejian Huang

**Affiliations:** Research Institute of Tropical Forestry, Chinese Academy of Forestry, Guangzhou 510520, China; sunlin721@163.com (L.S.); lzr15927358929@126.com (Z.L.)

**Keywords:** *Lumnitzera littorea*, endangered, MaxEnt, climate change, potential suitable habitats

## Abstract

The survival status of *Lumnitzera littorea* is near threatened globally and critically endangered in China. Clarifying its global distribution pattern and its changing trends under different future climate models is of great significance for the protection and restoration of its endangered status. To build a model for this purpose, this study selected 73 actual distribution points of *Lumnitzera littorea* worldwide, combined with 12 environmental factors, and simulated its potential suitable habitats in six periods: the Last Interglacial (130,000–115,000 years ago), the Last Glacial Maximum (27,000–19,000 years ago), the Mid-Holocene (6000 years ago), the present (1970–2000), and the future 2050s (2041–2060) and 2070s (2061–2080). The results show that the optimal model parameter combination is the regularization multiplier RM = 4.0 and the feature combination FC (Feature class) = L (Linear) + Q (Quadratic) + P (Product). The MaxEnt model has a low omission rate and a more concise model structure. The AUC values in each period are between 0.981 and 0.985, indicating relatively high prediction accuracy. Min temperature of the coldest month, mean diurnal range, clay content, precipitation of the warmest quarter, and elevation are the dominant environmental factors affecting its distribution. The environmental conditions for min temperature of the coldest month at ≥19.6 °C, mean diurnal range at <7.66 °C, clay content at 34.14%, precipitation of the warmest quarter at ≥570.04 mm, and elevation at >1.39 m are conducive to *Lumnitzera littorea*’s survival and distribution. The global potential distribution areas are located along coasts. Starting from the paleoclimate, the plant’s distribution has gradually expanded, and its adaptability has gradually improved. In China, the range of potential highly suitable habitats is relatively narrow. Hainan Island is the core potential habitat, but there are fragmented areas in regions such as Guangdong, Guangxi, and Taiwan. The modern centroid of *Lumnitzera littorea* is located at (109.81° E, 2.56° N), and it will shift to (108.44° E, 3.22° N) in the later stage of the high-emission scenario (2070s (SSP585)). Under global warming trends, it has a tendency to migrate to higher latitudes. The development of the aquaculture industry and human deforestation has damaged the habitats of *Lumnitzera littorea*, and its population size has been sharply and continuously decreasing. The breeding and renewal system has collapsed, seed abortion and seedling establishment failure are common, and genetic variation is too scarce. This may indicate why *Lumnitzera littorea* is near threatened globally and critically endangered in China. Therefore, the protection and restoration strategies we propose are as follows: strengthen the legislative guarantee and law enforcement supervision of the native distribution areas of *Lumnitzera littorea*, expanding its population size outside the native environment, and explore measures to improve its seed germination rate, systematically collecting and introducing foreign germplasm resources to increase its genetic diversity.

## 1. Introduction

*Luminitzera littorea* belongs to the genus *Lumnitzera* of the family Combretaceae. It has high ecological, scientific research, medicinal, and economic value and is an important mangrove plant. It usually grows in the middle–high- or high-tide zone parallel to estuary banks and often coexists with other dominant mangrove plants to form small mangrove communities. Its developed root system can hold soil, effectively weakening the destructive power of typhoons and tsunamis, and ensure the safety of coastal residents. *Lumnitzera littorea* also shows good carbon sequestration ability in the global carbon cycle [1,2]. Its wood is hard and has excellent corrosion resistance, and it can be applied to furniture making, construction materials, ship building, and other fields. *Lumnitzera littorea* is also a marine medicinal plant, rich in bioactive compounds, mainly containing tannins, flavonoids, phenols, terpenoids, and their derivatives. The leaf extract has antibacterial and anti-inflammatory effects [3].

*Lumnitzera littorea* is a thermophilic species with a narrow distribution range. Worldwide, it is sporadically distributed in South Asian countries such as India and Sri Lanka, and in Southeast Asian regions including Myanmar, Thailand, and Malaysia. In China, its natural distribution areas are only in Tielugang, Sanya, and Xincungang, Lingshui, on Hainan Island [4,5,6,7]. However, recent field investigations and studies have shown that *Lumnitzera littorea* is on the verge of extinction in China [4,5]. It is recognized as near threatened (NT) in the IUCN Red List of Threatened Species, though it does not pose an extinction risk globally for the time being. It is also an endangered species in China according to the Ramsar Convention and a plant with extremely small populations according to the National Conservation Project Plan for Wild Plants with Extremely Small Populations (2011–2015). The List of National Key Protected Wild Plants lists it as a first-class protected wild plant in China [6,7,8]. Thus, we believe that systematically analyzing the impact of environmental driving factors on the suitable habitats of *Lumnitzera littorea* at a global scale will reveal its ecological adaptability differences in different regions, providing a theoretical basis for future conservation and restoration efforts.

Global climate change is accelerating the alteration of the structure and function of the Earth’s ecosystem, profoundly impacting biodiversity [9]. During the Last Interglacial (LIG), the global climate was warmer than the present, and the surface water temperature in the mid-latitude Pacific Ocean increased when the ice sheets expanded and the sea level dropped [10]. During the Last Glacial Maximum (LGM), due to climate factors such as a reduction in Northern Hemisphere summer insolation, a decrease in tropical Pacific sea surface temperature, and a reduction in atmospheric CO_2_ concentration, the global ice sheets expanded significantly, and the global temperature decreased [11]. Due to changes in the Earth’s orbit, the Northern Hemisphere summer during the Mid Holocene (MH) was warmer than in the current period, and the CO_2_ concentration level was lower [12]. In future climate scenario predictions, the SSP (Shared Socioeconomic Pathways) model describes the potential impacts of climate change under different socioeconomic conditions by setting different socioeconomic development paths. Under different scenarios, the global temperature will follow different warming trends [13]. Climate change can lead to dynamic changes in the geographical distribution of species, changes distribution types, and reduction in or even disappearance of suitable habitats [14]. The climate states of different historical periods not only demonstrate the long-term impacts of natural driving factors on the climate system but also provide an important model validation basis for current and future climate simulations. Therefore, understanding how climate change drives the alteration of species’ geographical distribution is of great significance for formulating biodiversity conservation strategies.

To address the challenges brought by climate change, ecologists widely apply Species Distribution Models (SDMs) to combine species occurrence data with factors such as bioclimate, soil, and terrain, thereby achieving goals such as species conservation, biogeographical research, and addressing climate change [15]. Among all the available species distribution models, the algorithm implemented by the MaxEnt model is the simplest. It estimates the target probability distribution by calculating the probability distribution of maximum entropy, processes data information based on only the presence of certain factors, and analyzes the distribution of species under current and future conditions. This model still has strong robustness even when the sample size is small, and it is widely applied in research such as species distribution prediction, habitat suitability assessment, and the impact of climate change on species distribution. It is an important analytical tool in ecology, conservation biology, and climate change research [16].

To systematically reveal the dynamic distribution changes and key influencing factors of *Lumnitzera littorea* under different climate scenarios, this study uses the global distribution data for *Lumnitzera littorea*, combined with terrestrial climate and ocean surface data, and uses the optimized MaxEnt model to create prediction model. It simulates the response of *Lumnitzera littorea* to climate factors from the Last Interglacial to the next 60 years and determines the environmental factors that play a dominant role in its distribution; the distribution of potential suitable habitats in different periods; and its suitability in different regions. We do so with the hope of solving the following scientific questions: What are the dominant environmental factors affecting the distribution of *Lumnitzera littorea*, and what are the suitable environmental conditions for it? What are the changes in the spatial patterns of its potential suitable habitats during the paleoclimate (Last Interglacial, Last Glacial Maximum, and Mid-Holocene), the present, and under two different scenarios in the 2050s and 2070s? In which regions are its potential highly suitable habitats mainly distributed in China? Finally, against the background of its global “Near Threatened” status, why is it in an endangered state in China, and how should scientific and effective protection and restoration strategies be formulated?

## 2. Results

### 2.1. Screening of Environmental Factors

Based on the distribution data of *Lumnitzera littorea* and 37 environmental factors, the “jackknife” test in the Maxent model was selected to evaluate the importance of environmental factors. Environmental factors with low contribution rates, low permutation importance, and high correlation were removed. The screening process follows the principle of “high contribution + low correlation + strong ecological significance”. This reduced model redundancy and instability and covered all types of environmental factors affecting the distribution of *Lumnitzera littorea*, ensuring a good balance between the statistical independence and ecological interpretability of the model input variables [17]. The key environmental factors included climatic factors, soil factors, ocean factors, and elevation factors, which were used for further model optimization (Table 1).

### 2.2. Best Parameters and Model Accuracy Testing

Combined with the feature class parameters and regularization multipliers in the Kuenm package, the model parameters were optimized. The results showed that the regularization multiplier of the optimal model was 4.0, and the feature combination was L + Q + P (linear + quadratic + product type). During the optimization process, the model’s performance with different parameter combinations was evaluated, and comprehensive screening was carried out considering three aspects: statistical significance, predictive ability, and model complexity. The AICc value of the optimized model decreased significantly. The smaller the AICc, the better the balance between the fitting effect and complexity. The omission rate of the model’s test points was 0, indicating that the prediction results had extremely high accuracy (Figure 1). Compared with the default parameter model, the optimal model showed better performance in fitting the data, had a lower omission rate, and had a more concise model structure, reducing the risk of overfitting. The optimized model could more realistically reflect the distribution pattern of *Lumnitzera littorea* under the environmental gradient.

### 2.3. Model Accuracy Evaluation

This study, based on a MaxEnt model constructed with optimized parameters, simulated and evaluated the potential distribution of *Lumnitzera littorea* in different historical and future periods. For each period, the model was repeated 10 times, and based on this, a Receiver Operating Characteristic curve (ROC curve) was drawn to evaluate its predictive performance. The ROC curve reflects the predictive performance of the model under different thresholds, and the AUC value (Area Under Curve) is used to measure the overall discriminant ability of the model. Under current climate conditions, the average AUC value predicted by the model for *Lumnitzera littorea* is 0.982 ± 0.002. The blue line (representing the results of 10 simulations) significantly coincides with the red line (representing the average curve), and is much higher than the black line (corresponding to the random prediction level), verifying the reliability of the model under the current climate background (Figure 2). The AUC values of the model in other periods are between 0.981 and 0.985, and the standard deviation does not exceed 0.002, indicating that the model has high accuracy and stability in predictions for different periods and can effectively identify the changing trends of *Lumnitzera littorea’s* suitable habitats.

### 2.4. Dominant Environmental Factors Affecting the Distribution of Lumnitzera littorea

By combining the contribution rate, permutation importance, and correlations between the factors, the dominant environmental factors affecting the modern potential distribution of *Lumnitzera littorea* were obtained (Figure 3 and Table 2). The total contribution rate reached 87.2%: min temperature of the coldest month (bio6), mean diurnal range (bio2), clay content (soil28), precipitation of the warmest quarter (bio18), and elevation (ele31). Among them, the contribution rate and permutation importance of the min temperature of the coldest month were the highest, at 52.4% and 52.1%, respectively.

### 2.5. Response of the Lumnitzera littorea’s Potential Distribution to Environmental Factors

According to the predicted existence probability of *Lumnitzera littorea* and the response curve of the dominant climate factors (Figure 4), combined with the classification of suitable habitats, the optimal range of environmental factors corresponding to the highly suitable habitats is as follows: min temperature of the coldest month (bio6) ≥19.6 °C, mean diurnal range (bio2) <7.66 °C, the clay content (soil28) = 34.14%, precipitation of the warmest quarter (bio18) ≥570.04 mm, and the elevation (ele31) >1.39 m. The existence probability of *Lumnitzera littorea* increases with the min temperature of the coldest month and precipitation of the warmest quarter. When its survival probability reaches the maximum, these values correspond to 23.77 °C and 2183.70 mm, respectively. The existence probability first increases and then decreases with increased elevation, and the optimal elevation is above 1.39 m. The existence probability is negatively correlated with mean diurnal range; when the diurnal range of the mean temperature is lower than 12.43 °C, it is more suitable for the survival of *Lumnitzera littorea*.

### 2.6. Potential Distribution of and Changes in Lumnitzera littorea Under Different Climate Scenarios

We conducted a visual analysis based on the prediction results of the MaxEnt model to determine the distribution of potential suitable habitats for *Lumnitzera littorea* in different periods. Wet then divided the potential suitable habitats into regions and analyzed the changes in the areas on different continents in each period.

#### 2.6.1. Global Potential Geographical Distribution of *Lumnitzera littorea* Under Current Climate Conditions

The actual distribution of *Lumnitzera littorea* is along the coastlines of the Indian Ocean and the western Pacific Ocean, covering southern Thailand, the Malay Peninsula, Indonesia, the Philippine Islands, Papua New Guinea, the Solomon Islands, and the coastal areas of Queensland and the Northern Territory of northern Australia. It is concentrated in tropical islands and low-latitude coastal zones, showing obvious regional concentration characteristics and coastal strip-shaped distribution (Figure 5).

On the whole, the MaxEnt model’s prediction results show that under current climate conditions, the potential suitable habitats of *Lumnitzera littorea* present a spatial pattern of regional aggregated distribution, mainly concentrated in tropical and subtropical coastal areas, covering multiple currently distributed countries. The highly suitable habitats are mainly distributed in the tropical and subtropical coastal areas within the 30° north and south latitudes. Typical regions include the coasts of Southeast Asia and South Asia, northern Oceania, and the eastern coast of Africa (Figure 6). This is generally consistent with the actual distribution area of *Lumnitzera littorea*, verifying the effectiveness of the model prediction.

#### 2.6.2. Global Potential Geographical Distribution of *Lumnitzera littorea* in Paleoclimate and Future Climate Conditions

The MaxEnt model’s prediction results indicate that, under the paleoclimate scenario, *Lumnitzera littorea* did not have many potential suitable habitats. This might be because during historical periods such as ice ages and interglacial periods—affected by global climate change and sea-level fluctuations—its suitable environment was limited, and it was unable to form a stable distribution area. Under the future climate scenario, its distribution area shows a slow expansion trend, and the grade of the suitable area shifts toward high-suitability habitats, particularly in Asia and Africa.

In the early stage of the low-emission scenario (2050s (SSP126)), the potential suitable habitat of *Lumnitzera littorea* increased compared with that of the present. The area of the lowly suitable habitat decreased, the area of the moderately suitable habitat was relatively stable, and the highly suitable habitat increased significantly (Figure 7a). The area of the lowly suitable habitat decreased by 92,300 km^2^, with the most significant reductions in South America and North America (Figure 8a); the area of the moderately suitable habitat changed little, with a slight reduction in Asia and small increases in Oceania and South America (Figure 8b); the area of the highly suitable habitat expanded significantly, increasing by 39.34% compared with the present (Figure 8c); and the total global suitable habitat area increased by 312,300 km^2^, with an increase rate of 6.70% (Figure 8d). The ecological suitability of *Lumnitzera littorea* was further enhanced globally, mainly reflected by the conversion of moderately and lowly suitable habitats into highly suitable ones. In the later stage of the low-emission scenario (2070s (SSP126)), the highly suitable habitats along the eastern coast of India, the coast of the Bay of Bengal, the Malay Peninsula, the coast of the Indonesian Islands, the Philippines, and the coast of Papua New Guinea further expanded compared with the previous stage, and the connectivity between small islands was stronger (Figure 7b). The area of the lowly suitable habitat increased by 232,600 km^2^ (Figure 8a), and the change in the area of the moderately suitable habitat showed certain regional heterogeneity. The moderately suitable habitat in Asia decreased the most, with a reduction of 743,900 km^2^; the areas in Africa and North America decreased by 215,300 km^2^ and 201,800 km^2^, respectively; and the areas of the moderately suitable habitats in South America and Oceania increased by 293,400 km^2^ and 83,100 km^2^, respectively (Figure 8b). The highly suitable habitat has expanded significantly, showing an expanding trend across multiple continents. Asia has the most concentrated growth of suitable habitats, with a new area of 506,600 km^2^, followed by Africa, South America, North America, and Oceania (Figure 8c). The total global suitable habitat increased by 306,100 km^2^ compared with the present, with an increase rate of 6.57% (Figure 8d).

In the early stage of the high-emission scenario (2050s (SSP585)), compared with the present, the potential suitable area of *Lumnitzera littorea* further expanded. The lowly suitable habitat and the moderately suitable habitat decreased, and the highly suitable habitats expanded significantly (Figure 7c). The areas of the lowly suitable habitat and moderately suitable habitat decreased by 202,200 km^2^ and 173,600 km^2^, respectively, both showing a significant reduction in Asia (Figure 8a,b); the area of the highly suitable habitat increased by 608,900 km^2^, with the most obvious increase in Asia (Figure 8c); the total global suitable habitat increased by 233,200 km^2^, with an increase rate of 5.00%, and the suitable range increased in a small area (Figure 8d). The internal structure of the suitable was redistributed, and the suitability of the original suitable area improved. In the later stage of the high-emission scenario (2070s (SSP585)), the global potential suitable habitat of *Lumnitzera littorea* continued to expand slightly compared with the previous stage, and the suitability significantly improved in the eastern part of Asia and the coastal areas of South Asia (such as the southeastern coast of China, the eastern coast of India, and the south of Bangladesh) (Figure 7d). The area of the lowly suitable habitats changed little overall, but increased in Asia (Figure 8a); the area of the moderately suitable habitat decreased, with a total reduction of 246,800 km^2^, mainly due to the conversion of the moderately suitable habitat in Asia to a highly suitable habitat under the warming background (Figure 8b); the area of the highly suitable habitat reached 2,314,000 km^2^, an increase of 73.04% compared with the present (Figure 8c); and the total global suitable habitat increased significantly compared with the present, with an increase rate of 9.48% (Figure 8d). In the later stage of the high-emission scenario, the potential suitable area of *Lumnitzera littorea* shows an expansion trend and is more coherent.

### 2.7. Centroid Changes in the Potential Suitable Habitats of Lumnitzera littorea

*Lumnitzera littorea* demonstrates a sensitive response to climate change under different emission scenarios. The overall centroid shows a northward migration trend, of which the speed and range are significantly enhanced under the high-emission scenario. Taking the contemporary period as the reference position, the centroid of *Lumnitzera littorea* is located in the sea area on the west side of Indonesia, with geographical coordinates at a (109.81° E, 2.56° N). Under the low-emission scenario, the distribution adjustment of the centroid changes little. It slowly migrates southwestward from the contemporary position to b (109.36° E, 2.50° N) in the early stage. The change range is small, and the migration path is gentle. In the later stage, the centroid further shifts northward to c (110.14° E, 2.85° N). Under the high-emission scenario, the centroid migration range is drastic, with obvious northward and westward shifts. In the early stage, the centroid moves from the contemporary position to the northeast to d (110.10° E, 2.66° N), and in the later stage, the centroid significantly shifts to e (108.44° E, 3.22° N) (Figure 9).

### 2.8. The Potential Geographical Distribution of Lumnitzera littorea Under Contemporary Climate in China

Under contemporary climate conditions, the potential suitable habitats of *Lumnitzera littorea* in China are predicted to be mainly concentrated in the coastal areas of South China and Taiwan, presenting an obvious coastal distribution pattern, with significant differences in the suitability levels (Figure 10). Hainan Province is the core suitable habitat for *Lumnitzera littorea*, with a high degree of suitability along the coast. The highly suitable habitats mainly cover the coastal areas of Wenchang City, Chengmai County, Danzhou City, Lingao County, Haikou City, Sanya City, Lingshui County, Qionghai City, Wanning City, etc. The remaining coastal areas are moderately suitable habitats. In Guangdong Province, the highly suitable habitats along the coast are mainly distributed in Xuwen County, Leizhou City, Wuchuan City in Zhanjiang City, and the coast of Zhuhai City. The moderately suitable habitats are mainly concentrated in the Pearl River Delta and the coast of western Guangdong, specifically, the southern part of Shenzhen City; Zhuhai City; Taishan City in Jiangmen City; Yangxi County; Yangdong County in Yangjiang City; Dianbai District; Huazhou City in Maoming City; the coast of Shantou City; Jieyang City; and Shanwei City. The lowly suitable habitats are in the Chaozhou area in eastern Guangdong. In Guangxi, the highly suitable habitat along the coast is mainly in Fangchenggang City; the moderately suitable habitats are distributed in Qinzhou City and Beihai City. In Taiwan, the highly suitable habitats are significantly concentrated in the coastal areas of Tainan City, Kaohsiung City, and Pingtung County on the southwest coast; the moderately suitable habitats are along the coast of Taichung City and Chiayi County.

## 3. Discussion

### 3.1. Influence of Environmental Factors on the Suitable Growth of Lumnitzera littorea

The distribution and growth of mangroves are affected by various factors, including climate, geology, salinity, hydrodynamics, seed dispersal strategies, storm intensity and frequency, and human activities [18]. The results of this study indicate that the min temperature of the coldest month and mean diurnal range are important environmental factors in the distribution of *Lumnitzera littorea*. The precipitation of the warmest quarter, clay content, and elevation also significantly impact its distribution.

Given their adaptation range because of temperature, mangrove plants are divided into thermophilic narrow-distribution species, thermophilic wide-distribution species, and low-temperature-resistant wide-distribution species; the corresponding lowest monthly average temperatures are greater than 20 °C, 12–16 °C, and less than 11 °C, respectively [19]. This study shows that the survival probability of *Lumnitzera littorea* increases with the min temperature of the coldest month. When the temperature reaches 23.77 °C, the survival probability can reach 0.769, which is consistent with the characteristics of *Lumnitzera littorea* as a thermophilic narrow-distributed species [6]. Elevation determines the critical line for mangrove afforestation, affecting species distribution. Differences in elevation change the tidal inundation frequency, leading to changes in soil nutrients and microbial communities [20,21,22]. Mangroves are flooded by tides and are usually located in the tide flat area above the mean sea level [23]. As the frequency of flooding increases, the microbial biomass in mangrove soil decreases, and the enzyme activity first increases and then decreases [22]. *Lumnitzera littorea* naturally grows in high-tide-zone mudflats with a relatively low tidal inundation frequency and less influence from tidal action. This study shows that when elevation is ≥1.39 m, the survival probability of *Lumnitzera littorea* is relatively high, and its suitability first increases and then decreases with the increased elevation. This is in line with the habitat characteristics of mangroves. The mean diurnal range refers to the difference between the maximum and minimum temperatures in a day, reflecting the amplitude of daily temperature changes. In a study of the suitable habitats of *Corydalis*, the average diurnal temperature range was considered one of the most important environmental factors in species distribution [17]. The results of our study show that *Lumnitzera littorea* grows close to the land edge and is relatively sensitive to temperature changes. The corresponding suitable average diurnal temperature range is 12.34 °C. Thus, a stable temperature environment is more suitable for its growth: the smaller the difference, the higher the probability of its survival. The clay content refers to the proportion of particles less than 2 μm. Soils with higher clay content can usually maintain a higher cation exchange capacity; have good water-holding and salt-buffering capacities; aggregate structure stability; and avoid root exposure caused by tidal erosion [24]. When salt invades, clay releases adsorbed Ca^2+^ to replace Na^+^, reducing salt damage [25]. The seeds of *Lumnitzera littorea* need a continuously moist environment for germination. The slow permeability of clay soil can prevent rapid water loss and maintain the humidity required for seed germination.

Given the MaxEnt model’s research findings on other mangroves, there are differences in the dominant environmental factors among these plants, but they all tend to live in a warm and humid environment. Temperature, precipitation, and sea surface temperature are the main environmental factors for *Kandelia candel*. It has a relatively strong resistance to low temperatures, and it grows better when precipitation in the warmest quarter is higher than 740.61 mm [26]. As a pioneer mangrove species, *Avicennia marina* has a wide ecological niche. Its distribution is mainly driven by three environmental factors: elevation, seawater salinity in the coldest season, and the highest temperature in the hottest month. It can adapt to relatively low tidal flat areas, and its suitable habitat can extend to below −0.84 m. It maintains a high survival probability in areas where the max temperature in the warmest month is as high as 36 °C [27]. *Bruguiera hainesii* is listed as a critically endangered (CR) species. The average diurnal temperature range is the most dominant environmental factor affecting its distribution, and it can survive in areas with a relatively stable climate [28].

### 3.2. Lumnitzera littorea Geographical Distribution Changes

The results show that the suitable habitats of *Lumnitzera littorea* are distributed in a strip along the coastline of China. In Southeast Asia and northern Oceania, the distribution is mainly in moderately and highly suitable habitats, while in North and South America and Africa, it is mainly in the moderately lowly suitable habitats. From the paleoclimate to the present and then future climate change periods, the distribution area of *Lumnitzera littorea* changes from non-existent to existent, its adaptability gradually increases, and its suitable habitats expand to high latitudes. The suitable area reaches its maximum in the late stage of the high emission scenario. Centroid migration is the systematic change in the geographical location of the ecological optimal distribution center of a species’ potential suitable habitats against the background of climate change [29]. In this context, Chen et al. [30] analyzed the data for 764 species globally and found that the biological community migrated about 16.9 km toward high latitudes every ten years on average. The migration rate of the species distribution centroid is positively correlated with the regional warming amplitude. If species exhibit response lags, they will face a higher risk of local extinction in the future. In this study, the spatial displacement trend of *Lumnitzera littorea*’s distribution is consistent with the research results for existing mangrove taxa. The potential distribution area of mangroves in China is continuously expanding northward under climate warming, and there is potential to establish populations of *Kandelia candel* in northern Zhejiang and Fujian [26,31]. *Lumnitzera littorea* is mostly distributed in the transition from the high-tide zone to the land edge. Its seeds can reach suitable habitats without long-distance floating, but it has a small diffusion radius. Because the exocarp of its fruit is highly lignified, the density of the fiber layer increases after absorbing water, and buoyancy decreases. Its seeds have no air sacs or cavities; thus, they do not float well and will settle prematurely [32]. To avoid a situation where the potential suitable habitat of *Lumnitzera littorea* has changed but the species cannot spread to it, a dynamic buffer zone or pre-protection zone can be set up in the frontier area where its suitability continues to increase, increasing the possibility of population establishment. The potential suitable habitats of *Lumnitzera littorea* are relatively narrow globally, its ecological niche is limited, and it is highly sensitive to environmental factors such as temperature and moisture. There are only sporadic highly suitable habitats in some regions. This further supports the classification of it as near threatened (NT) in the IUCN Red List of Threatened Species. However, our model prediction results show that there are also potential highly suitable habitats for *Lumnitzera littorea* in Hainan, Guangdong, Guangxi, and Taiwan in China. However, its existing natural population is extremely small, on the verge of extinction in the wild, with its global near threatened status and critically endangered status in China demonstrating a protection gap.

### 3.3. Protecting and Restoring Lumnitzera littorea in China

The endangerment of *Lumnitzera littorea* in China may be due to two factors: (1) Habitat Destruction: Owing to the development of aquaculture and human deforestation, its habitat has been damaged, and its population has continuously declined. Its natural distribution in China has always been in Tielugang, Sanya (109° E, 18° N), and Xincungang, Lingshui (110° E, 18° N), Hainan Island. Between them, the number of *Lumnitzera littorea* plants in Tielugang in Sanya has not changed much, while the number in Xincungang has been reduced by humans, sharply decreasing from 340 plants in 2006 to 2 plants in 2016 [7]. (2) Collapse of the Breeding and Renewal System: Seed abortion and seedling establishment failure are common, and genetic variation is too limited. The empty embryo rate of *Lumnitzera littorea* seeds is as high as 76.54%; nearly 50% of fruits and seedlings are eaten by pests; no seedlings grow in existing forests; and natural renewal is very difficult. Thus, it has essentially lost its natural reproductive ability. The genetic structure of *Lumnitzera littorea* is singular, with a high risk of inbreeding, weak dispersal, and adaptability. It has experienced a serious population bottleneck throughout history, and the genetic variation between different populations is limited. Indeed, the *Lumnitzera littorea* population in China belongs to the extremely genetically poor eastern LH3 (ELL) lineage. Furthermore, the genetic variation within populations is low, and communication between them is limited by geographical isolation, and inbreeding depression is a serious threat [33,34].

At present, the protection and restoration of *Lumnitzera littorea* face challenges, but there are also many opportunities. The results of this study show that against the background of climate warming, the adaptability of *Lumnitzera littorea* in China as a whole shows an upward trend. The coastal areas in the south, such as Guangdong, Guangxi, Hainan, and the west side of Taiwan, are highly suitable for its distribution, demonstrating a positive protection prospect. Considering the causes of *Lumnitzera littorea*’s endangerment in China, we offer the corresponding restoration strategies:(1)In response to habitat destruction, awareness of the endangered status of *Lumnitzera littorea* should be improved, legislative protection and law enforcement supervision in its native distribution areas should be strengthened, and the population size outside the native environment should be expanded. In line with the Regulations of the People’s Republic of China on the Protection of Wild Plants and the “14th Five-Year Plan” National Conservation and Restoration Plan for Wild Plants with Extremely Small Populations, local protection laws and regulations should be improved, and a targeted protection plan should be formulated. Destructive activities such as reclamation and development should also be strictly limited, and local governments should be encouraged to implement protection responsibilities according to laws and regulations. The highly suitable habitats of *Lumnitzera littorea* identified in this study, such as most areas of Guangdong, Guangxi, and Hainan, should be selected as priority sites. In the planting process, the spatial distribution characteristics and dominant environmental factors of suitable habitats should be comprehensively considered, and combined planting experiments involving “high-quality seeds + suitable areas” should be carried out. The min temperature of the coldest month and elevation conditions of planting sites should be evaluated to ensure that they meet the optimal ecological range. Measures such as PVC pipe protection [35] and seedling height determination should be taken to assist in improving the survival rate of seedlings to restore the population size outside the plant’s original environment.(2)In response to the collapse of *Lumnitzera littorea*’s breeding and renewal system, the widespread occurrence of seed abortion and failure of seedling establishment, and the excessive lack of genetic variation, measures to improve the seed germination rate should be explored. The plant’s germplasm resources in the wild in China should be systematically collected, and foreign germplasm resources should be introduced to increase its genetic diversity. Fixed-point marking and seed collection of existing healthy mother trees should be conducted. By removing empty embryos and screening healthy seeds, the acquisition rate of effective seeds can be improved, and a small nursery should be established for sowing and seedling cultivation. In the southeast coast of Hainan, which is a highly suitable habitat, gene-mixing population establishment should be carried out to strengthen population diversity, alleviate the genetic vulnerability caused by historical bottlenecks and high inbreeding, and cultivate a resilient restoration population to avoid large-scale seedling death caused by extreme cold damage [36]. The eastern LH3 (ELL)-type population should be introduced from the homologous areas of *Lumnitzera littorea*—such as the central–eastern Philippines, the Solomon Islands, and Papua New Guinea—and artificial introduction and assisted pollination experiments should be carried out to increase the gene flow within the population by establishing a mixed population [34]. Our results show that the min temperature of the coldest month is the dominant factor affecting the distribution of *Lumnitzera littorea*. Considering the potential stress of climate fluctuations, it is necessary to prioritize the screening of outstanding cold-tolerant individual plants from the populations growing in the northern margin. The tolerance of these plants should be verified through artificial chilling stress experiments to enhance their survival ability.

## 4. Materials and Methods

### 4.1. Research Area Scope and Distribution Location Data

Given the distribution status of *Lumnitzera littorea* in nature, the study area was selected to form a 50 km buffer zone toward the land side based on the global continental coastline to best fit the intertidal living environment of this species [31]. The distribution data were sourced from open databases, the Global Biodiversity Information Facility (GBIF, https://www.gbif.org, accessed on 17 March 2024) and the Chinese Virtual Herbarium (CVH, https://www.cvh.ac.cn/, accessed on 16 March 2024). The MaxEnt model was used to model the species distribution based on the distribution records of species occurrences. Cleaning up duplicate and biased data is of great significance for model construction; therefore, based on the resolution of environmental variables (2.5′, approximately 5 km), the collected data were screened according to the longitude and latitude coordinates of the distribution points to avoid those of the same species in the same space being too close, ensuring only one distribution data point within a range of 2.5′. As *Lumnitzera littorea* grows in the intertidal zone where the ocean and land transition, we manually adjusted its distribution coordinates to ensure that its distribution was within the 50 km buffer zone. The precision of the adjusted longitude and latitude was three decimal places. Finally, 73 global effective distribution points were obtained (Figure 4).

### 4.2. Environmental Variables and Handling

This study uses 37 environmental factors, consisting of 31 terrestrial environmental factors and 6 marine factors (Table 3). The terrestrial environmental factors include 19 bioclimatic variables from the WorldClim database (Worldclim, https://worldclim.org/, accessed on 10 May 2024), 11 soil factors from the Harmonized World Soil Database (HWSD, https://www.fao.org/soils-portal/, accessed on 13 May 2024), and elevation data from the Global Terrain Model (ETOPO Global Relief Model 2022). The bioclimatic variables include six periods: the Last Interglacial (LIG), the Last Glacial Maximum (LGM), the Mid-Holocene (MH), the present, and the 2050s and 2070s. For future climate data, two Representative Concentration Pathways, SSP126 and SSP585, from the Shared Socio-economic Pathways (SSP) are selected, representing the most conservative (low) and the most wasteful (high) emission pathways, respectively. The soil factors include soil available water content, soil electrical conductivity, soil exchangeable sodium salt content, sulfate content, carbonate content, exchangeable bases, soil cation exchange capacity, pH, clay content, silt content, and organic carbon content. The ocean factors include six surface ocean bioclimatic factors from the World Ocean Atlas 2023 of the National Centers for Environmental Information, USA (NCEI, https://www.ncei.noaa.gov/, accessed on 8 June 2024): average temperature of surface ocean water, seawater salinity, seawater dissolved oxygen content, seawater phosphate content, seawater silicate content, and nitrate content data. Environmental factors were imported into ArcGIS10.8, the coordinate system WGS1984 was selected, and resampling and clipping tools were used to unify the resolution and range of the environmental factors to 2.5′ and a 50 km buffer zone.

### 4.3. Environmental Factor Screening and Model Prediction

When the correlation coefficient, |r|, of two variables is greater than 0.8, this usually indicates strong collinearity. To reduce the multicollinearity effect among environmental factors, the correlation between 37 environmental factors is calculated to identify redundant variables and simplify the environmental factor dataset for model construction. This can avoid overfitting model caused by the multicollinearity of environmental factors, which may lead to a large deviation in final output results, negatively impact the MaxEnt model’s prediction results, and affect its generalization ability and prediction accuracy. At the same time, the jackknife method is used to analyze and output the contribution rates of different environmental factors. Finally, environmental factors with higher contribution rates and permutation importance and an absolute value of correlation not exceeding 0.8 are selected for model construction.

### 4.4. Model Construction and Parameter Optimization

The Kuenm package (https://github.com/marlonecobos/kuenm, accessed on 9 August 2024) of R version 4.3.1 was used to optimize the regularization multiplier (RM) and feature class (FC) parameters of the model. The values of the regularization multiplier ranged from 0 to 4, increasing at intervals of 0.5. Combining 31 combinations formed by 5 types of feature class parameters, namely, linear (L), quadratic (Q), product (P), threshold (T), and hinge (H), a comprehensive evaluation of 248 potential models was carried out. Based on parameters such as statistical significance (partial ROC), prediction ability (low omission rates), and complexity (AICc values), the final model should be statistically significant, have a low omission rate, and have a delta AICc less than 2 to ensure simplicity and prediction accuracy.

### 4.5. Suitable Area Division and Centroid Migration Analysis

Using the natural break classification method, the distribution area of *Lumnitzera littorea* is divide into 4 levels: high-suitable habitat areas (FI > 0.451), medium-suitable habitat areas (0.451 ≥ FI > 0.216), low-suitable habitat areas (0.216 ≥ FI > 0.061), and unsuitable habitat areas (0.061 ≥ FI > 0). According to the MaxEnt model’s prediction results, we used ArcGIS10.8 to classify and color the suitability of *Lumnitzera littorea* in different regions and generate global suitability distribution maps for different periods. We identified suitable habitats of different grades and obtained the predicted area of suitable habitats in various periods. We combined with the global suitable habitat prediction map, determined the area of each continent by region, and determined the proportion of each continent with respect to suitable habitats of different grades. We combined with the actual distribution area; selected potential suitable habitats in the regions of Asia, Africa, and Oceania; and used the “Centroid Changes (Lines)” tool in the SDM toolbox of ArcGIS10.8 to conduct centroid migration analysis.

## 5. Conclusions

This study used the MaxEnt model to predict the potential suitable distribution habitats of *Lumnitzera littorea* globally, identified the dominant environmental factors affecting its distribution and the diffusion and migration trends of the community, further analyzed the reasons why it is near threatened globally but critically endangered in China, and provided a theoretical basis for its protection. The main conclusions are as follows:(1)The dominant environmental factors affecting the distribution of *Lumnitzera littorea* are min temperature of the coldest month, mean diurnal range, clay content, precipitation of the warmest quarter, and elevation. The suitable range indicates that *Lumnitzera littorea* prefers a warm and humid climate and a beach environment with a relatively high terrain.(2)The results of the suitable habitat prediction show that currently, Asia and Africa are the main regions where the potential suitable habitats of *Lumnitzera littorea* are distributed. Compared with the modern distribution, there were no potential suitable habitats during the paleoclimate period. Under future scenarios, the area of suitable habitats will increase, the proportion of highly suitable habitats will increase, and the distribution will be more continuous. The potential highly suitable habitats of *Lumnitzera littorea* in China are relatively narrow. Hainan Island is the core potential highly suitable habitat area, and there are fragmented highly suitable habitat areas in coastal areas such as Guangdong, Guangxi, and Taiwan. Under the trend of global warming, *Lumnitzera littorea* has a tendency to migrate to higher latitudes, and its adaptability will further improve with climate warming.(3)Under the influence of the development of the aquaculture industry and human deforestation, the habitat of *Lumnitzera littorea* has been damaged, and the population size has been continuously declining. At the same time, the self-breeding and renewal system has collapsed, and the genetic variation is extremely scarce. This may be the reason why it is near threatened globally and critically endangered in China. The endangerment level and restoration difficulty of *Lumnitzera littorea* are relatively high. It is necessary to enhance awareness of its endangered status, improve the legislative system, and strictly protect its native environment; establish a dynamic monitoring platform, combine highly suitable habitats with the stable mangrove ecosystem, and actively construct new planting areas; carry out artificial seedling cultivation and tissue culture to save its breeding and renewal systems; introduce foreign homologous germplasm resources to expand the genetic basis; and enhance the population adaptability, so as to effectively reverse its critically endangered status.

## Figures and Tables

**Figure 1 plants-14-02792-f001:**
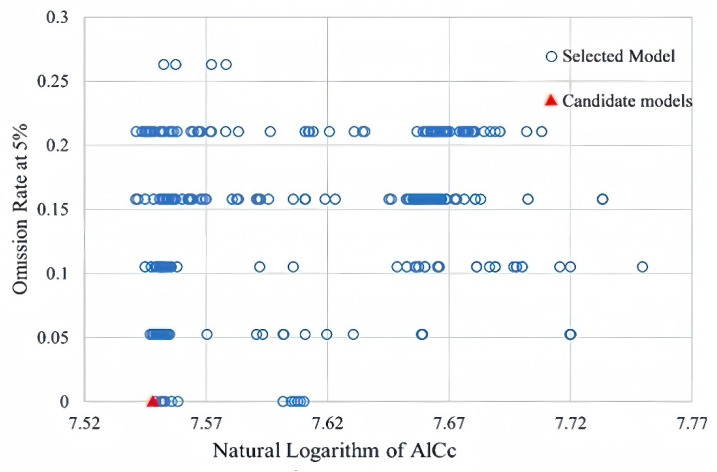
Model parameter selection results.

**Figure 2 plants-14-02792-f002:**
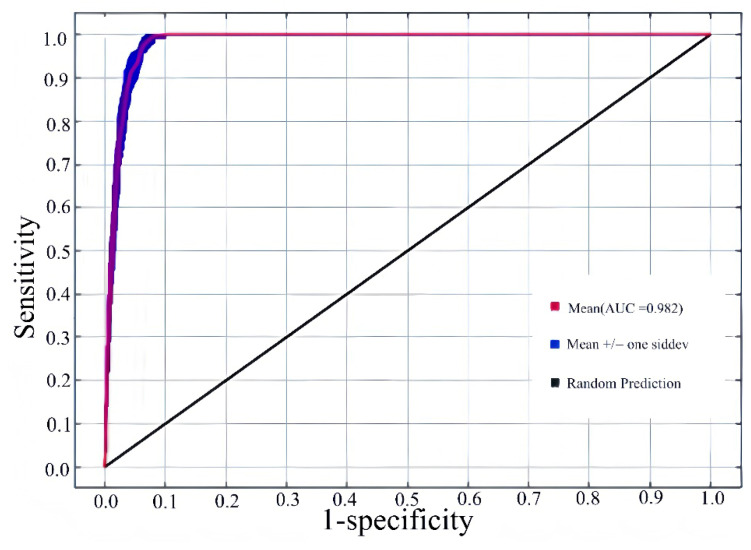
Receiver Operating Characteristic curve predicting the current potential suitable habitats of *Lumnitzera littorea*.

**Figure 3 plants-14-02792-f003:**
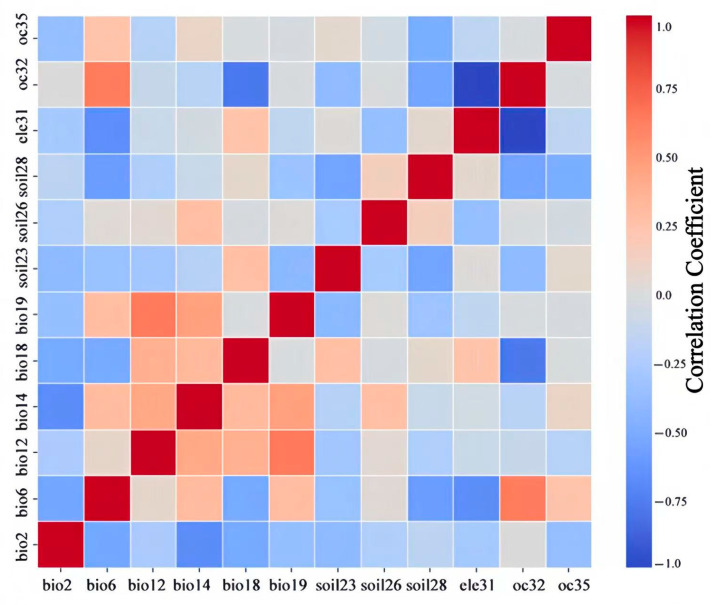
Spatial autocorrelation of environmental factors used to build the model. Note: The deeper the red color, the stronger the positive correlation. The deeper the blue color, the stronger the negative correlation.

**Figure 4 plants-14-02792-f004:**
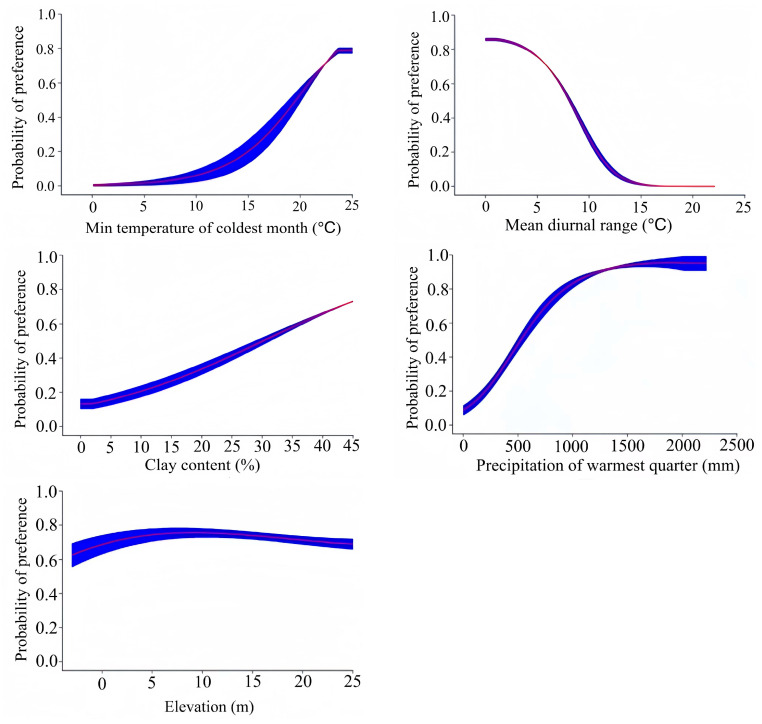
Response curves of the dominant environmental factors affecting the potential distribution of *Lumnitzera littorea*.

**Figure 5 plants-14-02792-f005:**
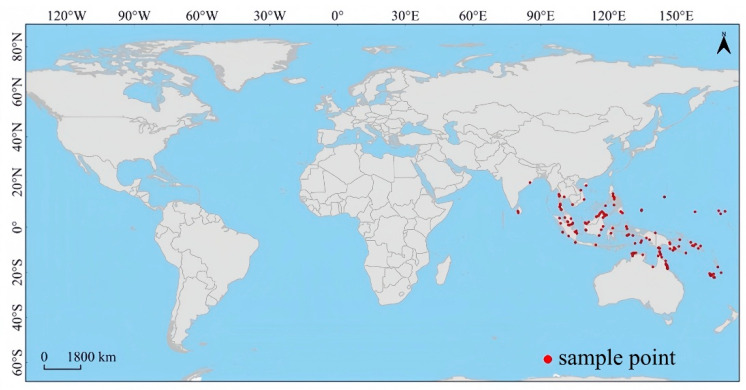
Global occurrence records of *Lumnitzera littorea*. Note: This map was created based on the standard map from map review No. GS (2021)648; the base map was not modified (same below).

**Figure 6 plants-14-02792-f006:**
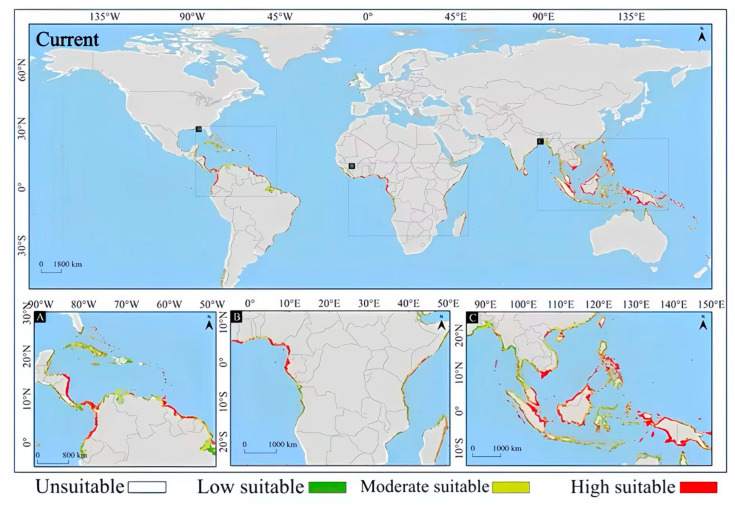
Prediction of the global potential suitable habitats of *Lumnitzera littorea* under current climate conditions. Note: A, B, and C in the figure are respectively the detailed diagrams of the selected parts from the global suitable habitat prediction maps above them.

**Figure 7 plants-14-02792-f007:**
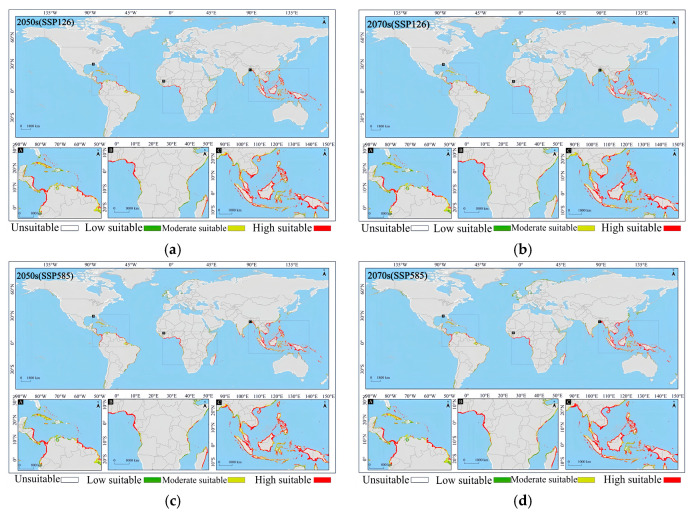
Prediction of the global potential suitable habitats of *Lumnitzera littorea* under future climates. Note: (**a**) 2050s (SSP126); (**b**) 2070s (SSP126); (**c**) 2050s (SSP585); (**d**) 2070s (SSP585). Note: A, B, and C in the figure are respectively the detailed diagrams of the selected parts from the global suitable habitat prediction maps above them.

**Figure 8 plants-14-02792-f008:**
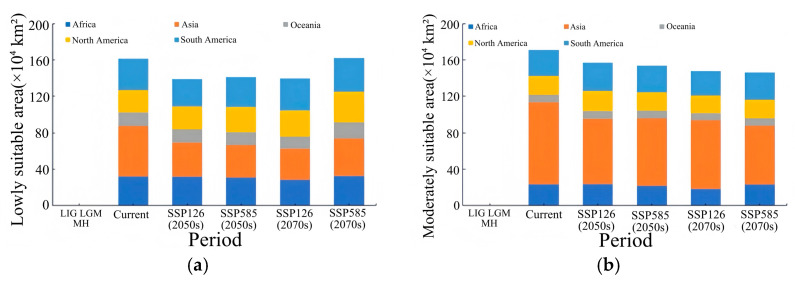
Predicted area of suitable habitats for *Lumnitzera littorea* in different periods. Note: (**a**) Lowly suitable habitats. (**b**) Moderately suitable habitats. (**c**) Highly suitable habitats. (**d**) Total suitable habitats.

**Figure 9 plants-14-02792-f009:**
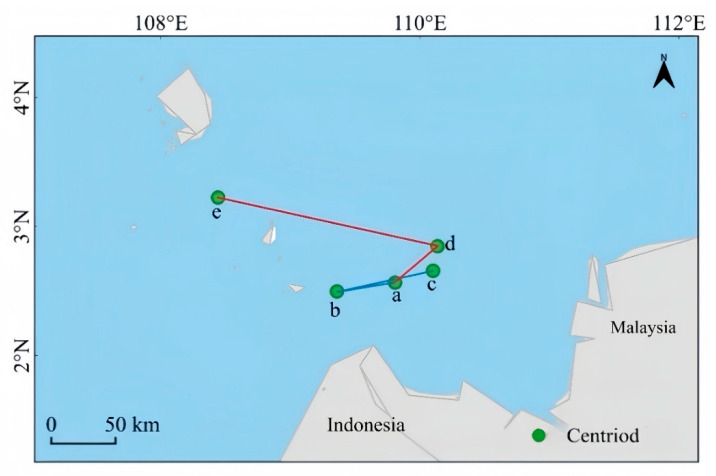
The centroid migration of potential suitable habitats of *Lumnitzera littorea.* Note: (a) current; (b) 2050s (SSP126); (c) 2070s (SSP126); (d) 2050s(SSP585); (e) 2070s (SSP585).

**Figure 10 plants-14-02792-f010:**
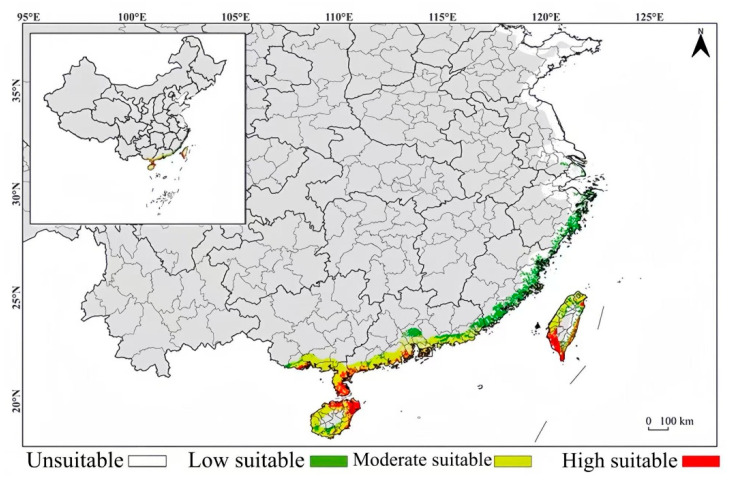
Prediction of the current potential suitable habitats of *Lumnitzera littorea* in China.

**Table 1 plants-14-02792-t001:** The environmental factors used for model construction.

	Code	Environmental Factors and Descriptions
Climate factors	bio2	Mean diurnal range (mean of monthly (max temp–min temp))
	bio6	Min temperature of coldest month
	bio12	Annual precipitation
	bio14	Precipitation of driest month
	bio18	Precipitation of warmest quarter
	bio19	Precipitation of coldest quarter
	bio23	Sulfate content
	bio26	Soil cation exchange capacity
Soil factors	soil28	Clay content
Terrain factors	ele31	Elevation
Ocean factors	oc32	Surface ocean average temperature
	oc35	Phosphate content in seawater

**Table 2 plants-14-02792-t002:** The contribution rates and permutation importance of dominant environmental factors affecting the potential distribution of *Lumnitzera littorea*.

Code	Percent Contribution/%	Permutation Importance/%
bio6	52.4	52.1
bio2	23.1	31.2
soil28	7.2	2.1
bio18	2.3	1.5
ele31	2.2	9.9

**Table 3 plants-14-02792-t003:** The environmental factors used to analyze potential suitable habitats.

	Code	Environmental Factors and Descriptions	Unit
Climate factors	bio1	Annual mean temperature	°C
	bio2	Mean diurnal range (Mean of monthly (max temp–min temp))	°C
	bio3	Isothermality (bio2/bio7) (×100)	-
	bio4	Temperature seasonality (standard deviation × 100)	-
	bio5	Max temperature of warmest month	°C
	bio6	Min temperature of coldest month	°C
	bio7	Temperature annual range (bio5–bio6)	°C
	bio8	Mean temperature of wettest quarter	°C
	bio9	Mean temperature of driest quarter	°C
	bio10	Mean temperature of warmest quarter	°C
	bio11	Mean temperature of coldest quarter	°C
	bio12	Annual precipitation	mm
	bio13	Precipitation of wettest month	mm
	bio14	Precipitation of driest month	mm
	bio15	Precipitation seasonality (coefficient of variation)	-
	bio16	Precipitation of wettest quarter	mm
	bio17	Precipitation of driest quarter	mm
	bio18	Precipitation of the warmest quarter	mm
	bio19	Precipitation of coldest quarter	mm
Soil factors	soil20	Soil available water content	mm
	soil21	Soil electrical conductivity	dS/m
	soil22	Soil exchangeable sodium salts	%
	soil23	Sulfate content	%
	soil24	Carbonate or lime content	%
	soil25	Exchangeable bases	cmol/kg
	soil26	Soil cation exchange capacity	cmol/kg
	soil27	pH	-
	soil28	Clay content	%
	soil29	Silt content	%
	soil30	Organic carbon content	%
Terrain factors	ele31	Elevation	-
Ocean factors	oc32	Surface ocean average temperature	°C
	oc33	Seawater salinity	%
	oc34	Dissolved oxygen in seawater	umol/kg
	oc35	Phosphate content in seawater	umol/kg
	oc36	Silicate content in seawater	umol/kg
	oc37	Nitrate content in seawater	umol/kg

## Data Availability

The original contributions presented in this study are included in the article. Further inquiries can be directed to the corresponding author.

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
