# Peer review of "A Study on the Endangerment of Luminitzera littorea (Jack) Voigt in China Based on Its Global Potential Suitable Areas"

_plants, 2025, doi:10.3390/plants14172792_

Round 1

Reviewer 1 Report

Comments and Suggestions for Authors

This manuscript presents a comprehensive study using MaxEnt modeling to predict suitable habitats for the endangered mangrove species Lumnitzera littorea across different temporal scales and climate scenarios. The work addresses an important conservation question and provides valuable insights into the species' ecological requirements and potential distribution changes.

Major Comments

  1. Data Quality and Sample Size

The use of 73 global distribution points appears adequate for MaxEnt modeling, but the authors should discuss potential sampling bias, particularly the concentration in certain geographical regions.

The manual adjustment of coordinates to fit within the 50-km buffer zone needs better justification and potential impact on model accuracy should be discussed.

  1. Environmental Variable Selection

While the authors screened 37 environmental factors down to 12, the selection criteria could be more transparent. The statement about "high contribution + low correlation + strong ecological significance" needs quantitative thresholds.

The correlation threshold of |r| > 0.8 is reasonable, but the authors should provide a correlation matrix or supplementary table showing the final selected variables' correlations.

  1. Model Validation

The model shows excellent performance metrics, but lacks independent validation with withheld data or different time periods.

Cross-validation results or spatial block validation would strengthen the reliability assessment.

Minor Comments

  1. Figure 1 (model parameter selection) is difficult to interpret. Consider adding clearer labels and legends.
  2. Only two SSP scenarios (126 and 585) were used. Including intermediate scenarios (e.g., SSP245) would provide a more complete picture.
  3. Line 18: "feature combination FC = L + Q + P" - define abbreviations on first use
  4. Line 131: "Jackknife" should be "jackknife" (not a proper noun)
  5. Table 1: Consider reorganizing by factor type (climate, soil, ocean, terrain) for better readability
  6. Line 447: "Tielugang of Sanya and Xincungang of Lingshui" - provide coordinates for international readers
  7. References: Several Chinese references lack English translations of titles
  8. Figure 3 (response curves) could benefit from confidence intervals or uncertainty bands.
  9. Standard deviations for AUC values are very small (≤0.002), suggesting possible overfitting or insufficient variation in model runs.
  10. The natural break classification method for habitat suitability levels needs better justification compared to other threshold selection methods.
  11. Line 44: "increase its genetic diversity" - be more specific about proposed methods
  12. Lines 425-441: The discussion of seed dispersal limitations could be better integrated with the modeling results
  13. The term "paleoclimate scenario" is used inconsistently - specify which historical periods are referred to
  14. The 2.5' resolution (~5 km) may be too coarse for coastal species habitat modeling, especially in complex coastal environments.
  15. Six marine variables were included but their relative importance is not well discussed. Given the coastal/marine nature of mangroves, this deserves more attention.

This study makes a valuable contribution to mangrove conservation science by providing the first comprehensive habitat suitability analysis for Lumnitzera littorea. The methodology is generally sound, and the results have clear conservation implications. The identified environmental thresholds and future habitat projections will be valuable for conservation planning. However, the manuscript would benefit from addressing the methodological concerns raised above, particularly regarding model validation and uncertainty assessment. The work successfully explains why L. littorea is critically endangered in China despite being only near-threatened globally, and provides a solid foundation for evidence-based conservation strategies. With minor revisions addressing the technical and clarity issues noted above, this manuscript will make a strong contribution to the literature.

Author Response

Thank you for your careful review. You can refer to the submitted file for specific modifications.

Reviewer 2 Report

Comments and Suggestions for Authors

The article is interesting in that, by applying Maxent's modeling, they reconstruct the potential distribution of Lumintzera littorea under various current and future climate scenarios.
However, I note important critical issues that will need to be resolved before the article can be published in the journal. In particular, considering that the species' actual distribution is concentrated in Southeast Asia as far north as Australia, it makes little sense to consider its potential distribution in continents like Africa or Central America, where the species is not present and never has been. The authors, however, discuss this in the same manner as for Asia, where the species exists. The biogeographical aspect must also be taken into account!
Given that the authors' intention is to focus on its potential in China in order to initiate programs to protect and spread the species, it is important to indicate the populations actually present in China on the distribution maps. Now they are lacking.
Furthermore, the authors announce their intention to consider the species' distribution in past periods, such as the Last Glacial Maximum, but there is no trace of this in the results or discussion. Therefore, they must decide whether to address the topic, possibly supported by the presence of fossils (e.g., wood or pollen), in the areas indicated by the model, or not consider it and therefore eliminate it from the introductory preamble.

Finally, many statements in the text should be better supported by relevant citations, which are currently quite few.

For these reasons, it is recommended that the article be carefully reviewed and the topics and critical issues highlighted be addressed more thoroughly.

Author Response

(The authors gave the same response as above.)

Round 2

Reviewer 1 Report

Comments and Suggestions for Authors

The authors have thoroughly addressed all concerns from the initial review, significantly enhancing the manuscript's clarity, structure, and scientific rigor. The introduction clearly highlights the importance of studying Lumnitzera littorea's global distribution and its conservation implications, particularly given its critically endangered status in China. The methodology is well-detailed, with a clear explanation of the MaxEnt model parameters (RM = 4.0, FC = L + Q + P) and the integration of 73 global distribution points with 12 environmental factors across six time periods (Last Interglacial, Last Glacial Maximum, Mid-Holocene, present, 2050s, and 2070s). The model’s high predictive accuracy (AUC values of 0.981–0.985), low omission rate, and concise structure address previous concerns about validation. Key environmental factors, such as minimum temperature of the coldest month (≥19.6°C), mean diurnal range, clay content, precipitation, and elevation, are well-articulated, providing insight into the species’ ecological requirements. The analysis of centroid shifts under the SSP585 scenario and the identification of core suitable habitats in Hainan, Guangdong, Guangxi, and Taiwan are robust and insightful. The proposed conservation strategies—strengthening legislative protection, expanding populations, and enhancing seed germination and genetic diversity—are practical and well-aligned with the study’s findings, effectively linking threats like aquaculture, deforestation, seed abortion, and low genetic variation to the species’ endangered status. The manuscript is now comprehensive, readable, and scientifically sound, making a valuable contribution to mangrove conservation. I recommend accepting the manuscript for publication.

Author Response

Dear Reviewer,

Hello! Thank you very much for taking the time out of your busy schedule to review the revised article. We appreciate your positive and constructive comments on this article, which have been of great help to us in further improving it. In accordance with your suggestions, we have made corresponding revisions and improvements to the article, and the quality of the article has been significantly enhanced. At the same time, we are grateful for your recognition of this article, which gives it the opportunity to be published in the journal. Finally, we would like to thank you again for the time and energy you have devoted to reviewing this article. We wish you every success in your work and all the best!

Best regards,

Ms. Lin Sun et al.

Reviewer 2 Report

Comments and Suggestions for Authors

The text has been significantly improved, and the authors have taken into account most of the suggestions.

However, an unresolved issue remains because they discuss a global distribution around the world of the species, but  it is essentially distributed in Southeast Asia. It is unlikely that it will spread to areas such as Africa and Central America. Therefore, the authors must address this issue and, if necessary, highlight that the species' ecological potential does not correspond to its actual distribution  unless it will be introduced as an alien species into areas such as Africa and Central America. A minimum discussion on this topic should be developed.

Author Response

Dear reviewer,

Hello, thank you very much for taking the time out of your busy schedule to review this revised manuscript. Regarding your feedback, we would like to respond as follows: This study discusses the global potential suitable habitat distribution of Lumnitzera littorea. According to the model prediction results, the potential suitable habitats of Lumnitzera littorea are concentrated in Southeast Asia, which is basically consistent with its actual distribution. You mentioned that this species is unlikely to spread to regions such as Africa and Central America, but the prediction results indicate that there are also suitable habitats for it in Africa. This suggests that Lumnitzera littorea may survive in African regions in the future, and its suitable habitats are likely to expand. This study is precisely a prediction of the potential suitable habitats of Lumnitzera littorea; the absence of its distribution in these regions now does not mean it will definitely not survive there in the future. We hope this explanation is satisfactory to you. Thank you again for the time and effort you have devoted to reviewing this article. We wish you every success in your work and all the best!

Best regards,

Ms. Lin Sun et al.